# Virologic suppression among HIV-positive pregnant and lactating women receiving antiretroviral therapy in Africa: A systematic review and meta-analysis

Berihun Agegn Mengistie[1]*, Getie Mihret Aragaw[1], Gebrye Gizaw Mulatu[2], Kindu Yinges Wondie[3], Alemneh Tadesse Kassie[3], Alemken Eyayu Abuhay[4], Wondimnew Mersha Biset[5], Moges Tesfa Tsega[6], Abay Eyayu Asrie[7], Tazeb Alemu Anteneh[3], Habtu Kifle Negash[8], Eshet Gebrie[9], Nuhamin Tesfa Tsega[10]

1 Department of General Midwifery, School of Midwifery, College of Medicine and Health Sciences, University of Gondar, Gondar, Ethiopia, 2 Department of Health Informatics, Institute of Public Health, College of Medicine and Health Sciences, University of Gondar, Gondar, Ethiopia, 3 Department of Clinical Midwifery, School of Midwifery, College on Medicine and Health Sciences, University of Gondar, Gondar, Ethiopia, 4 University of Gondar Comprehensive Specialized Hospital, Gondar, Ethiopia, 5 Department of Anesthesiology, Critical Care and Pain Medicine, Saint Paul's Hospital Millennium Medical College, Addis Ababa, Ethiopia, 6 Department of Internal Medicine, School of Medicine, College of Medicine and Health Sciences, University of Gondar, Gondar, Ethiopia, 7 Department of Obstetrics and Gynecology, School of Medicine, College of Medicine and Health Sciences, University of Gondar, Gondar, Ethiopia, 8 Department of Human Anatomy, School of Medicine, College of Medicine and Health Sciences, University of Gondar, Gondar, Ethiopia, 9 Department of Clinical Chemistry, School of Biomedical and Laboratory Sciences, College of Medicine and Health Sciences, University of Gondar, Gondar, Ethiopia, 10 Department of Women's and Family Health, School of Midwifery, College on Medicine and Health Sciences, University of Gondar, Gondar, Ethiopia

* berihunagegn21@gmail.com

## Abstract

### Background

The elimination of mother-to-child transmission of human immunodeficiency virus (HIV) is a key global public health priority. In Africa, virologic failure among people living with HIV continues to pose a significant public health challenge, affecting both individual well-being and community health. Maintaining viral load suppression is crucial to prevent vertical transmission of HIV and to minimize maternal morbidity and mortality. To stop the vertical transmission of HIV and lower the risk of maternal morbidity and mortality, it is important to achieve viral load suppression. Although many African countries have adopted the global 95-95-95 targets, comprehensive data on virologic suppression among pregnant and lactating mothers across the continent remains limited. The objective of this systematic review and meta-analysis was to determine the pooled estimate of virologic suppression and to examine the factors associated with it among HIV-positive pregnant and lactating women on antiretroviral therapy in Africa.

**Data availability statement:** All relevant data are within the manuscript and its Supporting Information files.

**Funding:** The author(s) received no specific funding for this work.

**Competing interests:** The authors have declared that no competing interests exist.

**Abbreviations:** ART, Antiretroviral therapy; AOR, Adjusted odds ratio; CI, Confidence Interval; CD4, Cluster of differentiation 4; HIV, Human Immunodeficiency Virus; JBI, Joanna Briggs Institute; MTCT, Mother-to-Child Transmission of HIV; SRM, Systematic review and meta-analysis; VL, Viral load; VS, Virologic Suppression; UNAIDS, United Nations Programme on HIV/AIDS; WHO, World health organization.

## Methods

This study followed the Preferred Reporting Items for Systematic Reviews and Meta-Analyses (PRISMA) guidelines. The study protocol was registered in the International Prospective Register of Systematic Reviews (PROSPERO; CRD420251186121). We carried out a thorough systematic review by examining PubMed, ScienceDirect, Hinari, and Google Scholar for relevant studies. Data from the studies were retrieved using an Excel sheet and analyzed with STATA version 17. The Joanna Briggs Institute appraisal tool was used to evaluate the methodological quality of studies. A random-effects model with restricted maximum likelihood (REML) was applied to determine the pooled prevalence of virologic suppression (viral load threshold ≤1000 copies/ml) among pregnant and lactating mothers in Africa. A funnel plot and the Egger's test were used to investigate publication bias. Statistical heterogeneity was assessed using the $I^2$ statistic and Cochrane's Q test.

## Results

A total of 55 eligible studies, comprising 304,883 participants, were included in the quantitative meta-analysis. Accordingly, the overall prevalence of virologic suppression among HIV-positive pregnant and breastfeeding women in Africa was 80.86% (95% CI: 77.63%, 84.09%, $I^2$ = 99.84%). In contrast, the pooled estimate for achieving an undetectable viral load was substantially lower, at 60.92% (95% CI: 52.46%, 69.39%; $I^2$ = 99.91%). Virologic suppression was significantly associated with women's age (15–24 years) (AOR = 0.49; 95% CI: 0.32–0.77), disclosure of HIV status to a partner (AOR = 1.66; 95% CI: 1.31–2.11), first-line antiretroviral therapy regimen (AOR = 6.53; 95% CI: 1.93–22.06), and good antiretroviral drug adherence (AOR = 3.61; 95% CI: 1.18–11.02). In addition, other socio-demographic variables, higher educational level, being married/cohabitant, urban residency, healthcare utilization (time of ANC booking, time of ART initiation, duration of ART), fear of stigma, distance to health facility, shortage of health professionals, ART drug stock-out, and lack of HIV care commodities were significantly associated with virologic suppression among HIV-positive pregnant and lactating women in Africa.

## Conclusion

The pooled estimate of virologic suppression among HIV-positive pregnant and breastfeeding women in Africa was approximately 81%, below the global target of 95% virological suppression. This emphasizes the necessity of targeted strategies for younger HIV-positive women, disclosing HIV status, initiating first-line antiretroviral regimens, and promoting antiretroviral treatment adherence. Upgrading health care systems to enable regular viral load monitoring, as well as addressing socio-demographic and antiretroviral therapy-related variables, are vital steps towards attaining and sustaining VS in these groups of population, ultimately assisting in achieving elimination of MTCT of HIV.

## Introduction

Human immunodeficiency virus (HIV) is one of the world's most critical public health challenges, affecting millions of people [1,2]. According to the Joint United Nations Programme on HIV/AIDS (UNAIDS) and World Health Organization (WHO) report, approximately 40.8 million people worldwide were infected with HIV, of which 21.0 million were women (15 + years); 1.3 million new HIV cases were reported, and 630,000 people died from AIDS-related illnesses at the end of 2024. In this report, nearly 1.4 million children aged below 15 years were living with HIV [1,2]. Sub-Saharan African countries were responsible for 65% (an estimated 26.3 million) of the worldwide HIV burden [1,2]. Women and girls of all ages contributed to 44% of all new HIV infections globally in 2023 [3].

The primary route of HIV infection in children is mother-to-child transmission (MTCT) of HIV during pregnancy, childbirth, and postpartum periods through breastfeeding [4]. It accounts for more than 90% of all newly acquired pediatric HIV infections worldwide [5]. Without intervention, the overall MTCT of HIV incidence ranges from 15% to 45% [6]. However, appropriate interventions during pregnancy, labor, and breastfeeding can lower the risk of perinatal HIV transmission to less than 2% in non-breastfeeding and 5% in breastfeeding populations, respectively [7,8].

The World Health Organization (WHO) is committed to a global public health mission that focuses on expanding access to HIV testing, antiretroviral therapy (ART), and treatment monitoring, aiming to enhance clinical management of HIV, achieve sustained viral suppression, and lower HIV-related morbidity and mortality [9]. Maternal plasma HIV viral load (VL) status is the strongest determinant of HIV transmission from mother to child (MTCT) [10–12]. Nowadays, viral load assessment has become the main approach for monitoring HIV patients' clinical and virologic outcome to ART [13]. HIV viral load is categorized into three main stages: unsuppressed (>1000 copies/mL), suppressed (detected but ≤1000 copies/mL), and undetectable (<50 copies/mL or not detected by the assay used) [9,10,14]. In 2021, the World Health Organization revised its HIV treatment monitoring algorithm to enhance support for people living with HIV in attaining viral suppression, with the ultimate aim of maintaining an undetectable viral load [9]. Achieving HIV viral suppression is crucial for enhancing individual health, preventing sexual transmission, and lowering the risk of perinatal transmission [9]. On the contrary, pregnant and lactating women who do not achieve virologic suppression (VS) are at a higher risk of HIV-related complications, death, and vertical transmission to their infants [10,15].

Studies show that achieving and maintaining VS among pregnant and postpartum WLWH remains to be challenging, despite significant progress in the scaling up of ART via prevention of mother-to-child transmission (PMTCT) initiatives [16–18]. Several countries have been implementing the PMTCT program packages [19–21]. It includes lifetime ART for pregnant and breastfeeding women, preventive antiretroviral (ARV) medications for HIV-exposed infants (HEIs), serial testing for HEIs, and prompt ART initiation for children who acquire HIV [4,22]. Maintaining virologic suppression (≤1000 copies/ml) in HIV-positive pregnant and postpartum women is a global public health goal for improving maternal health and producing HIV-free children [12,23].

Antiretroviral therapy (ART) as part of PMTCT is an important strategy to combat the global AIDS crisis and lower vertical transmission rates [24]. The WHO guidelines indicated that ART should be initiated in all pregnant and breastfeeding HIV-positive women, regardless of their VL, clinical stage, or CD4 cell count [15]. The fixed-dose combination (FDC) of tenofovir disoproxil fumarate (TDF), lamivudine (3TC), and dolutegravir (DTG) is the ideal first-line regimen for the treatment of HIV infection in adults and adolescents, including pregnant and breastfeeding women [25]. Timely initiation and consistent adherence to ART are key predictors for achieving virologic suppression, specifically undetectable HIV viral load and preventing mother-to-child transmission of HIV, mitigating a range of clinical, virological, and immunological risks [15,26]. Additionally, viral suppression enhances health outcomes for HIV-positive women, notably decreased illness, longer life expectancy, and better birth outcomes [27–29].

To tackle the continuing HIV crisis, the UNAIDS program is striding toward a 95-95-95 global target, ensuring that 95% of individuals living with HIV are aware of their HIV status, 95% of those diagnosed initiate lifelong ART, and at least 95%

of patients beginning ART achieve virologic suppression [30–32]. However, in many of the SSA countries, virologic suppression is still behind the UNAIDS 95-95-95 targets [32–34].

In Africa, ART treatment failure or virologic failure among HIV-positive individuals continues to be a significant public health concern [35–37]. Similarly, the rate of virologic suppression among pregnant and lactating women is still unsatisfactory throughout the continent [16,38]. In Africa, previous study findings on the proportion of virologic suppression among pregnant and breastfeeding women living with HIV revealed inconsistent and equivocal results that ranged from 29.1% to 97.8% [39,40]. A body of evidence has shown that sociodemographic variables, adherence to ART, ART regimens, enhanced counseling, baseline viral load status, duration of ART, disclosing HIV status, medication tolerance, drug–drug interactions, CD4 cell counts, prior treatment history, and antiretroviral drug resistance were found to be significantly associated with virologic suppression [36,41–44].

Achieving the third target of the UNAIDS 95-95-95 goals, ensuring that 95% of people living with HIV on ART achieve sustained VS, is essential to curbing HIV transmission and improving health outcomes [45]. Despite substantial progress in scaling up ART and PMTCT services through national, regional, and international initiatives, virologic suppression among pregnant and breastfeeding women remains suboptimal in many African countries. This enduring gap represents a critical, yet under-addressed challenge with profound implications for maternal health and the prevention of mother-to-child HIV transmission. This persistent gap highlights an urgent, unresolved challenge with critical implications for both maternal health and the prevention of mother-to-child transmission of HIV. Numerous studies have demonstrated that late ART initiation, poor adherence during pregnancy and the postpartum period, insufficient viral load monitoring, treatment interruptions, loss to follow-up during breastfeeding, and delayed ART initiation are all interrelated factors that contribute to virologic non-suppression. Sociocultural and institutional obstacles, including stigma, limited access to high-quality maternal HIV services, socioeconomic vulnerability, and limitations in the health system, further hamper long-term participation in maternity care [44–47,48].

Despite the fact that numerous studies document local viral suppression rates in particular countries, there is no rigorous and consolidated quantitative synthesis for this particular population group (pregnant and lactating women) has been conducted across Africa. This systematic review and meta-analysis addresses this gap by pooling data from diverse settings, estimating overall virologic suppression rates, and identifying key determinants of virologic suppression in these particular population groups. This study is unique in its geographical scope and target population since it consolidates fragmented and heterogeneous findings into a coherent continent-wide perspective. It provides robust and actionable evidence to guide health professionals, policymakers, and guideline developers in designing and implementing targeted, evidence-based strategies to enhance virologic suppression in these populations, ultimately contributing to eliminating perinatal transmission of HIV.

## Methods and materials

### Study protocol and search strategy

This systematic review adhered to the Preferred Reporting Items for Systematic Reviews and Meta-Analysis (PRISMA) standards [49]. The study protocol for the review has been registered on PROSPERO with reference ID CRD420251186121 and available on: https://www.crd.york.ac.uk/PROSPERO/view/CRD420251186121 (**S1 File**).

A rigorous and comprehensive review of Google Scholar, PubMed, Hinari, and ScienceDirect for original research on virologic suppression and associated factors conducted in Africa. The inclusion and exclusion criteria were established based on the condition, context, and study population (Co Co Pop) approach. Previous studies that met the eligibility criteria were retrieved.

A search strategy was developed for the databases by combining keywords with Boolean operators. Ultimately, we employed the following search term combination: "viral load suppression", "virologic suppression", "viral suppression", "HIV viral load non-suppression", "viral load", "detectable viral load", "undetectable viral load", "virological suppression",

"HIV viral load", "plasma viremia", "HIV viral suppression", "HIV viremia", "HIV-positive women", pregnant, "lactating mothers", "postpartum women", "breastfeeding mothers", "antiretroviral therapy", ART and Africa. To obtain additional articles from the citation list of publications found in the databases that were accessible, snowballing techniques were also employed (S2 File).

## Eligibility criteria

**Inclusion criteria.** Condition: The condition of interest was virologic suppression among HIV-positive pregnant and lactating women receiving antiretroviral therapy in Africa.

Context: All primary studies that reported the prevalence and/or associated factors of virologic suppression in the Africa context.

Population: The study population for this review is pregnant and lactating mothers living with HIV on ART.

Study design: All primary observational studies, including cross-sectional, case-control, and cohort studies that reported the prevalence and/or associated factors virologic suppression among HIV-positive pregnant and lactating women receiving ART.

Publication year: Both published and unpublished articles from end of 2015 (benchmark for the beginning of the Sustainable Development Goals (SDG) era, and July 30, 2025, were included.

**Exclusion criteria.** Articles that failed to report the outcome of interest, narrative reviews, qualitative reviews, expert comments, case reports, editorials, letters, and methodological studies were excluded from the review.

## Measurement of outcome variables

This study aimed to determine the pooled prevalence of virologic suppression among pregnant and lactating women receiving antiretroviral therapy in Africa. This study's second objective was to determine the variables that are associated with virologic suppression in these key populations. The World Health Organization (WHO) defined virological suppression as when viral load copies become ≤1,000 copies/mL of blood after six months of ART initiation [9,50]. However, virologic failure (VF) is defined as a VL persistently over 1000 copies/mL in two consecutive measurements after 6 months (3 months apart) on enhanced ART adherence support. In this systematic review and meta-analysis, we employed a viral load threshold of ≤1,000 copies/mL (virologic suppression), which is consistent with the WHO guideline criteria utilized in standard clinical decision-making [16,50,51].

## Quality assessment for included studies

The quality of the included primary studies was evaluated independently by two authors (BAM and NTT). Each study was assessed using the Critical Appraisal Checklist developed by the Joanna Briggs Institute (JBI). Twenty-eight of the included papers were evaluated using the 9-item cross-sectional checklist, while 27 used the 11-item cohort checklist [52,53]. Studies with a JBI checklist score of five or above were regarded as being of acceptable quality, indicating a low likelihood of bias. Any disputes between the two authors were settled by open conversation and consultation with GMA, the third author.

## Data extraction and management

Data extraction from the included publications was performed independently by two authors (BAM and NTT) using a standardized abstraction form developed in Excel. In accordance with the inclusion and exclusion criteria, all identified studies were imported into the reference manager EndNote 20. After screening titles and abstracts, full texts were reviewed for eligibility. Disagreements were resolved through discussion with a third reviewer (GMA). To avoid duplication, studies with overlapping or identical data were excluded. For eligible studies, the following information was extracted: identification

 

details (first author's last name and year of publication), proportion of virologic suppression, factors associated with virologic suppression, adjusted odds ratios with 95% confidence intervals, study area, study design, study population, sample size, and risk of bias assessment method (**S3 File**, **S4 Table**).

### Data synthesis and statistical analysis

All statistical data analyses were carried out after the data were collected using a Microsoft Excel spreadsheet and exported to the statistical program STATA version 17. Texts, tables, and forest plots were used to display the extracted data. A binomial distribution was used to determine each study's standard error of prevalence of the outcome variable. The Higgins I-squared ($I^2$) test was used to check for heterogeneity in the pooled prevalence of the studies. The heterogeneity among the included studies was classified as low, moderate, or high based on $I^2$ values of <25%, 50–75%, and >75%, respectively [54].

A random-effects model with restricted maximum likelihood (REML) was used to determine the pooled prevalence of virologic suppression in Africa. REML is preferred over the Der Simonian and Laird method because it employs a likelihood-based approach, which typically offers a more stable and unbiased estimate of between-study variance ($\tau^2$) and more accurate confidence interval coverage, especially in the presence of high heterogeneity. Even though Der Simonian and Laird's method offers a narrower confidence interval (CI), it underestimates the between-study variance ($\tau^2$) and inflates type I error (false positive) under extreme heterogeneity, which affects the robustness of the pooled estimate [55,56].

A subgroup analysis and meta-regression analysis were conducted across the publication year, country, regions of Africa, ART regimen, and study population to identify potential sources of study heterogeneity. In addition, we performed a leave-one-out sensitivity analysis to look at how particular studies influence the pooled estimate. The pooled estimates from across the continent were then displayed in forest plots and tables, along with their corresponding 95% confidence intervals. Potential small-study effects were assessed using funnel plots and Egger's regression test. Visually, publication bias has been assessed using a funnel plot (23). Statistically, potential publication bias was evaluated using Egger's test, with a p-value < 0.05 suggesting evidence of publication bias [57]. Finally, we conducted a meta-analysis and systematic synthesis to examine the determinants of virologic suppression among pregnant and lactating women receiving ART in Africa.

## Results

### Study selection and characteristics of the included studies

All searched databases, including Google Scholar, yielded a total of 1,639 research articles; 406 duplicate records were deleted, and the remaining 1,233 papers were further screened. However, the majority of papers (n = 1,108) were removed after reading their titles and abstracts. The remaining 125 full-text articles were then reviewed for eligibility criteria, and 70 studies were excluded for a variety of reasons, including insufficient data, a difference in the viral load suppression cutoff point, variation in the study context, being unrelated to the outcome of interest, systematic reviews, and qualitative reviews. Then, the remaining 55 eligible studies with a total of 304,883 HIV-positive pregnant and breastfeeding women were included in the final quantitative meta-analysis [16,28,44,51,58–79,80–83,84–94]. Among the included studies, 26 articles had reported an undetectable VL threshold (VL < 50 copies/ml). In terms of study distribution across Africa, 31 and 13 studies were undertaken in Southern Africa and Eastern Africa, respectively (Fig 1).

### Prevalence of virological suppression among pregnant and lactating mothers receiving ART in Africa

In this meta-analysis, the pooled prevalence of virologic suppression among pregnant and lactating women receiving ART in Africa was 80.86% (95% CI: 77.63%, 84.09%, $I^2$ = 99.84%, $\tau^2$ = 146.21, p-value = 0.000). Additionally, the pooled estimate

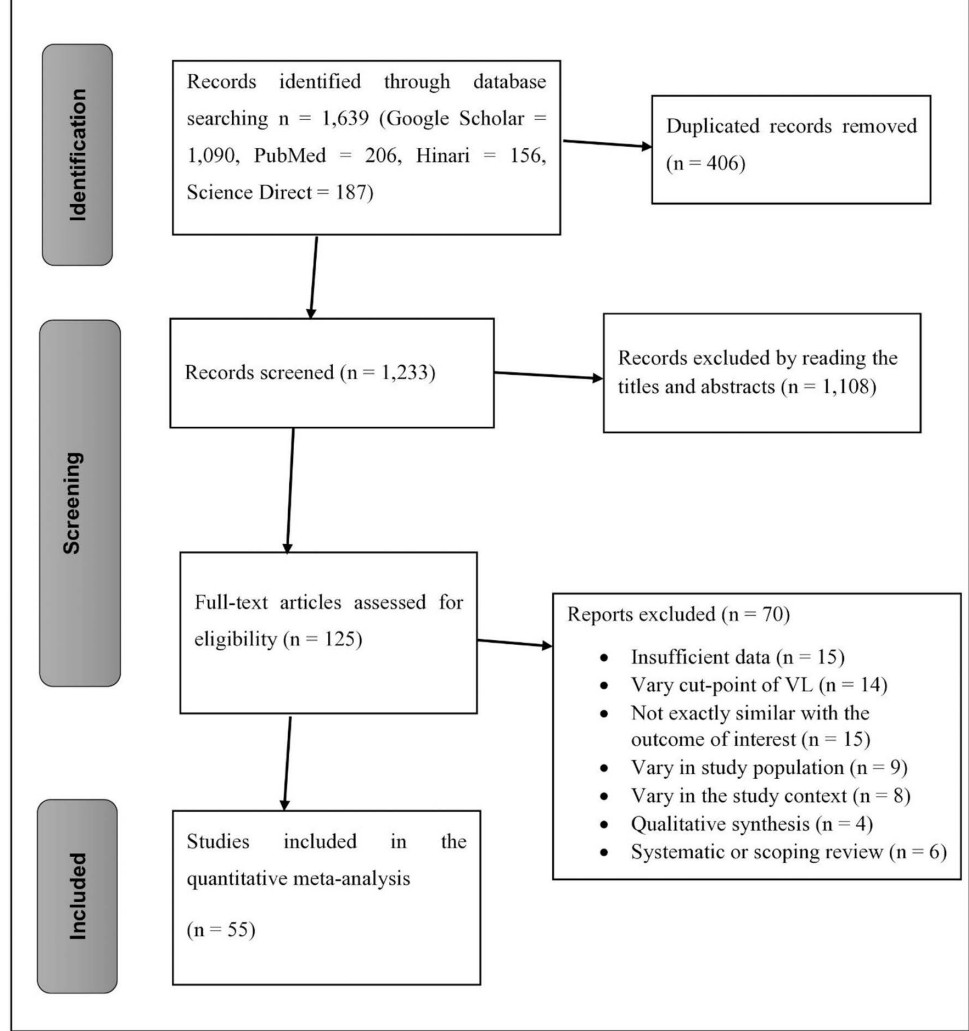

**Fig 1. PRISMA flow diagram showing the selection process of studies for the virologic suppression among HIV-positive pregnant and lactating women receiving ART in Africa.**

showed that only 60.92% (95% CI: 52.46%, 69.39%, $I^2 = 99.91\%$) of participants achieved an undetectable viral load status. Subgroup analyses were conducted to explore potential sources of heterogeneity, such as differences in study setting, population characteristics, and study design. Besides, sensitivity analyses were performed to assess the robustness of the pooled prevalence and to determine whether any single study had a disproportionate influence on the overall results (Fig 2).

The 95% prediction interval (PI) was 57.4% to 95.5%, indicating that the effect size of a new study conducted in a comparable setting could plausibly fall within this CI. Even though the pooled prevalence offers a useful overall summary, the wide prediction interval indicates substantial between-study heterogeneity. Accordingly, the pooled estimate should be interpreted as a descriptive summary of existing studies rather than a precise estimate for all settings. Due to the substantial heterogeneity among studies, a median-based summary was employed as an alternative statistical method. Consequently, the median prevalence across studies was 84.1% (IQR: 75.5%, 89.7%), providing a robust, non-parametric summary that is not influenced by extreme values or high variability among studies.

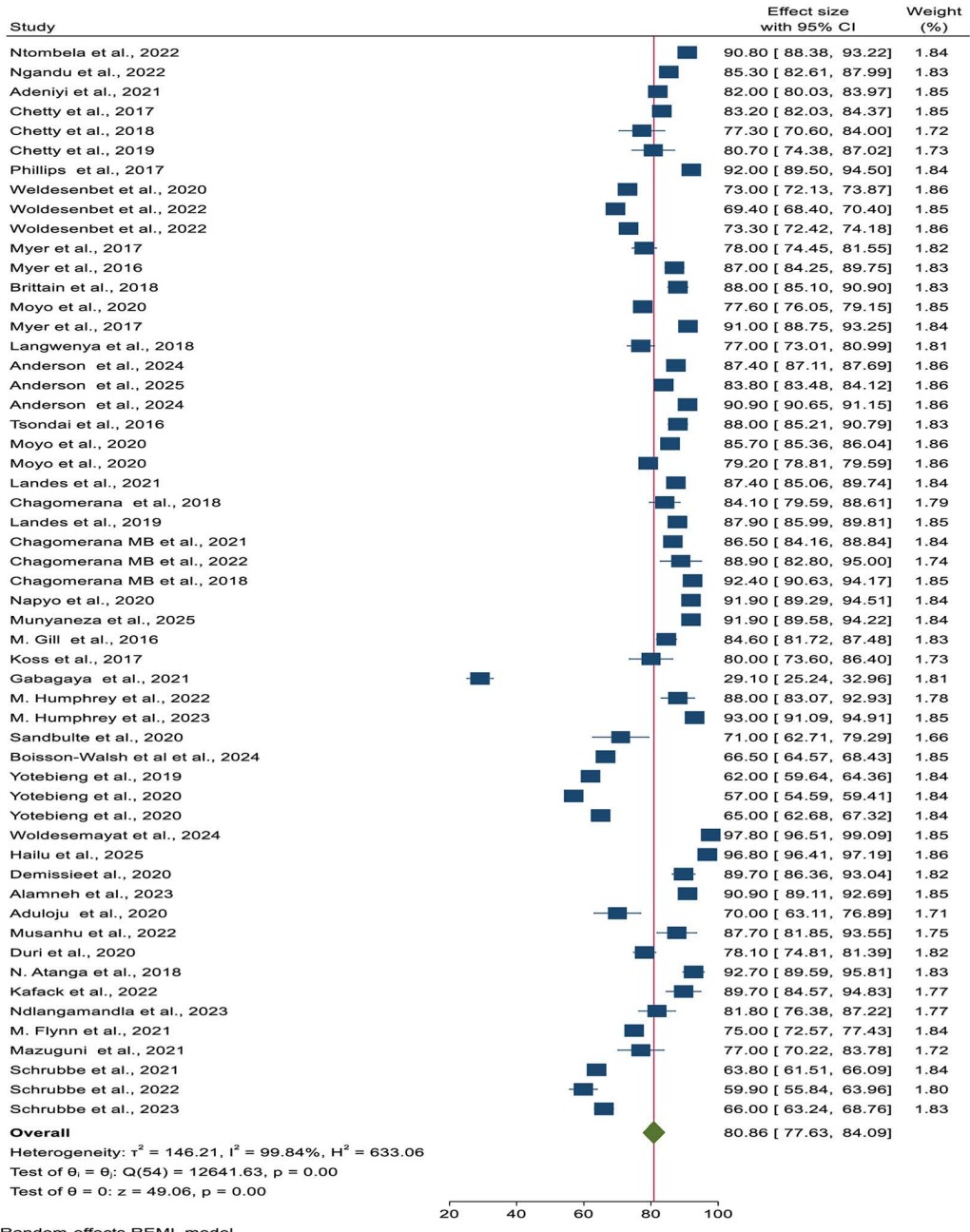

**Fig 2. A forest plot that shows the pooled prevalence of virologic suppression among HIV-positive pregnant and lactating women receiving antiretroviral therapy in Africa.**

## Heterogeneity and subgroup analysis

Subgroup analysis was performed based on the country, the regions of Africa, and the study population. Accordingly, Ethiopia had the highest prevalence of VS, 94.11% (95% CI: 91.16, 97.07), while the Democratic Republic of Congo had the lowest prevalence, 62.66% (95% CI: 58.56, 66.75). Moreover, the lowest overall prevalence of VS in pregnant and

lactating mothers living with HIV was found in the Central Africa region (62.66%; 95% CI: 58.56, 66.75), whereas the highest was in Western and East Africa, which found nearly 84% of viral suppression. In terms of the study population, 81% of pregnant and postpartum mothers had VS, which is nearly the same pooled prevalence.

Overall, the subgroup analysis demonstrated substantial geographic variation in virologic suppression across Africa, highlighting differences in ART program effectiveness and context-specific variables. This significant between-studies heterogeneity suggests that multiple factors, such as variations in ART regimens, level of ART adherence, health system capacity, availability of viral load monitoring, and sociocultural barriers, contribute to the observed differences in virologic suppression (**Table 1**).

## Publication bias

The presence of publication bias was graphically checked using a funnel plot, and it was confirmed statistically using Egger's test. Thus, the funnel plot shows some visual asymmetry, but according to the Egger's test (p-value = 0.2402), there is no statistically significant evidence of publication bias. The asymmetry may be due to extreme heterogeneity across studies (Fig 3).

**Table 1. Subgroup analysis for virologic suppression among HIV-positive pregnant and lactating women receiving antiretroviral therapy in Africa.**

| Parameters | Studies | VS (%) at 95% CI | I² (%) | P-value |
|---|---|---|---|---|
| **Stud y design** | | | | |
| Cohort studies | 27 | 82.74% (80.80, 84.69) | 99.35% | 0.000 |
| Cross-sectional studies | 28 | 79.34% (74.06, 84.63) | 99.69% | 0.000 |
| **Country** | | | | |
| Cameroon | 2 | 91.89% (89.23, 94.55) | 0.00 | 0.327 |
| DR Congo | 4 | 62.66% (58.56, 66.75) | 92.49 | 0.000 |
| Ethiopia | 4 | 94.11% (91.16, 97.07) | 94.97 | 0.000 |
| Kenya | 3 | 84.82% (74.86, 94.78) | 92.80 | 0.000 |
| Malawi | 6 | 88.06% (85.62, 90.51) | 80.81 | 0.000 |
| SSA | 3 | 66.28% (60.04, 72.51) | 95.09 | 0.000 |
| South Africa | 21 | 82.8% (80.42, 85.21) | 99.64 | 0.000 |
| Uganda | 3 | 66.99% (23.93, 110.05) | 99.72 | 0.000 |
| Rwanda | 2 | 88.30% (81.15, 95.46) | 93.31 | 0.000 |
| Zimbabwe | 2 | 82.58% (73.20, 91.97) | 87.27 | 0.005 |
| Others* | 3 | 76.52% (69.68, 83.35) | 71.31 | 0.031 |
| **Regions of Africa** | | | | |
| Central Africa | 4 | 62.66% (58.56, 66.75) | 92.49 | 0.000 |
| East Africa | 13 | 83.49% (77.69, 89.3) | 99.09 | 0.000 |
| Sub-Saharan Africa | 4 | 66.28% (60.04, 72.51) | 95.09 | 0.000 |
| Southern Africa | 31 | 83.74% (81.72, 85.76) | 99.47 | 0.000 |
| Western Africa | 3 | 84.49% (72.77, 96.20) | 94.24 | 0.000 |
| **Study population** | | | | |
| Pregnant women | 38 | 81.15% (78.66, 83.63) | 99.61 | 0.000 |
| Postpartum women | 9 | 80.61% (73.28, 87.94) | 99.08 | 0.000 |
| Both | 8 | 80.21% (73.25, 87.18) | 98.87 | 0.000 |

*Nigeria, Estiwani, Tanzania.

                                                                                      

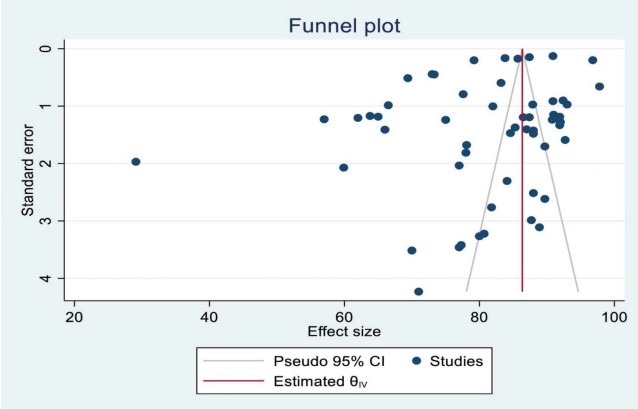

**Fig 3. A funnel plot test that indicates the virologic suppression among HIV-positive pregnant and lactating women receiving antiretroviral therapy in Africa.**

### Sensitivity analysis

To examine the impact of a single study on the estimated effect size, a leave-one-out sensitivity analysis was conducted using the random-effects model. The results, however, demonstrate that the pooled estimate was not substantially impacted by a single study, and the point estimate of the excluded study is within the confidence interval of the overall estimate of virologic suppression.

This demonstrated that the average estimate of virologic suppression among HIV-positive pregnant and breastfeeding mothers in Africa was robust (Fig 4).

**Meta-regression analysis.** A meta-regression analysis, together with subgroup and sensitivity analyses, was conducted to identify potential sources of heterogeneity. Study-level characteristics, including publication year (before vs. after 2020), ART regimen, country, African region, and sample size, were entered as covariates in the model. The analysis showed that use of a DTG-based ART regimen and the African region were significant moderators, accounting for part of the observed heterogeneity across studies. Thus, DTG-based ART regimens demonstrated significantly higher virologic suppression compared to non-DTG-based regimens ($\beta = 16.43$, $p = 0.012$). This suggests that variations in ART regimens had a significant role in the observed heterogeneity between studies, with DTG-based therapy significantly associated with virologic suppression. Furthermore, the region of Africa was substantially linked with viral load suppression ($\beta = -2.92$, $p = 0.037$), indicating considerable regional variations in virologic suppression rate. This study suggests that contextual factors related to geographic location, such as disparities in health system capability, implementation of ART programs, and VL monitoring, influence the observed between-study heterogeneity (**Table 2**).

### Factors associated with virologic suppression among pregnant and lactating mothers receiving ART in Africa

This systematic review and meta-analysis investigated 16 studies that identified factors associated with virologic suppression among pregnant and lactating mothers receiving ART in Africa [16,17,44,59,60,63,64,73,75,78,79,84,86,95]. In general, virologic suppression among HIV-positive pregnant and lactating women was substantially correlated with sociodemographic variables (women's age, higher educational level, being married/cohabitating, urban residency) [16,17,44,51,59,64,73,95–97], healthcare utilization (time of ANC booking and ART initiation, regimen and duration of ART, baseline VL threshold, adherence status of ART) [17,39,44,59,64,95,96,98], and accessibility and availability of healthcare services (long distance to health facility, shortage of health professionals, inadequate counseling, ART drugs running out, sub-optimal sample transportation) [12,99,100].

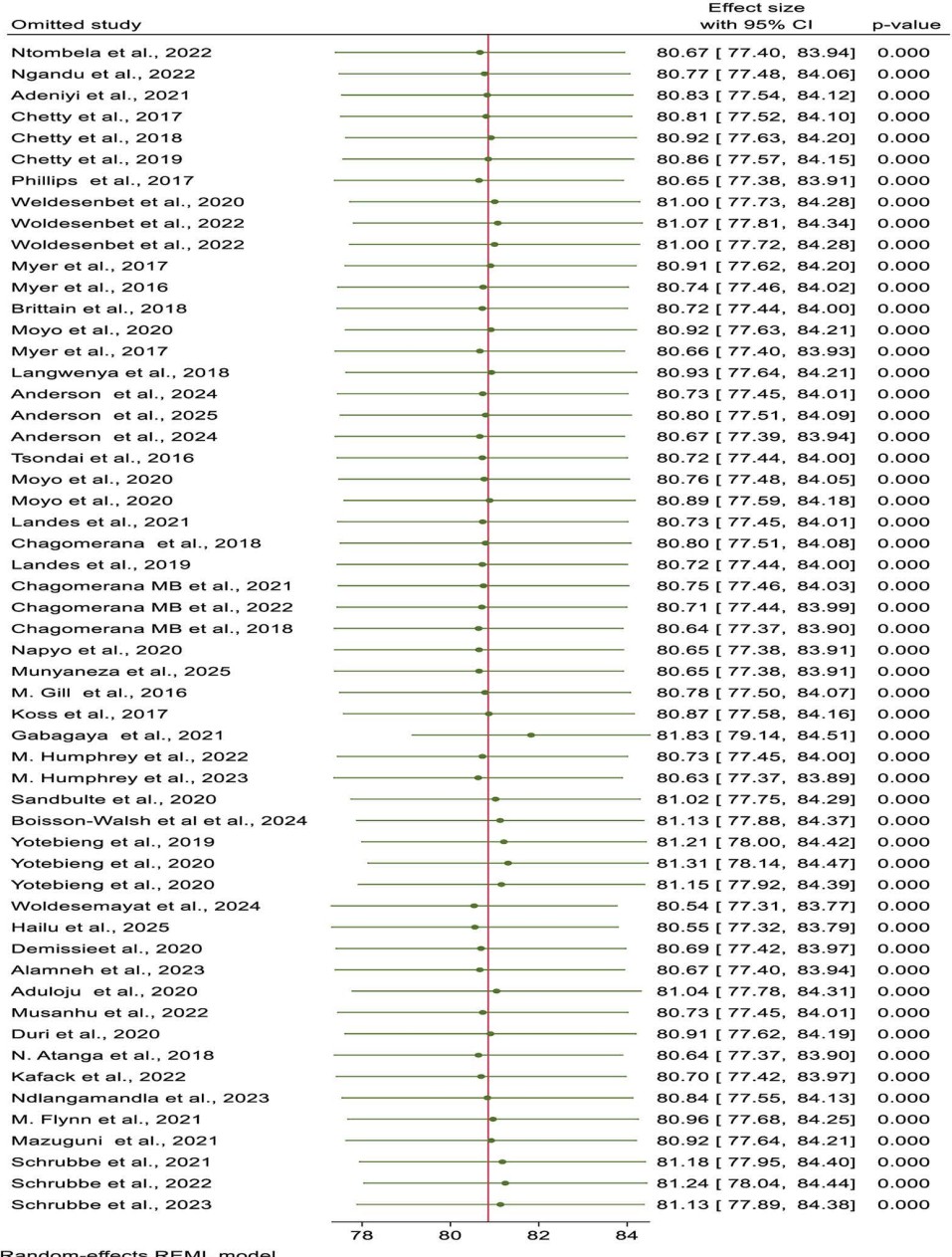

| Omitted study | | Effect size with 95% CI | p-value |
|---|---|---|---|
| Ntombela et al., 2022 | | 80.67 [ 77.40, 83.94] | 0.000 |
| Ngandu et al., 2022 | | 80.77 [ 77.48, 84.06] | 0.000 |
| Adeniyi et al., 2021 | | 80.83 [ 77.54, 84.12] | 0.000 |
| Chetty et al., 2017 | | 80.81 [ 77.52, 84.10] | 0.000 |
| Chetty et al., 2018 | | 80.92 [ 77.63, 84.20] | 0.000 |
| Chetty et al., 2019 | | 80.86 [ 77.57, 84.15] | 0.000 |
| Phillips et al., 2017 | | 80.65 [ 77.38, 83.91] | 0.000 |
| Weldesenbet et al., 2020 | | 81.00 [ 77.73, 84.28] | 0.000 |
| Woldesenbet et al., 2022 | | 81.07 [ 77.81, 84.34] | 0.000 |
| Woldesenbet et al., 2022 | | 81.00 [ 77.72, 84.28] | 0.000 |
| Myer et al., 2017 | | 80.91 [ 77.62, 84.20] | 0.000 |
| Myer et al., 2016 | | 80.74 [ 77.46, 84.02] | 0.000 |
| Brittain et al., 2018 | | 80.72 [ 77.44, 84.00] | 0.000 |
| Moyo et al., 2020 | | 80.92 [ 77.63, 84.21] | 0.000 |
| Myer et al., 2017 | | 80.66 [ 77.40, 83.93] | 0.000 |
| Langwenya et al., 2018 | | 80.93 [ 77.64, 84.21] | 0.000 |
| Anderson et al., 2024 | | 80.73 [ 77.45, 84.01] | 0.000 |
| Anderson et al., 2025 | | 80.80 [ 77.51, 84.09] | 0.000 |
| Anderson et al., 2024 | | 80.67 [ 77.39, 83.94] | 0.000 |
| Tsondai et al., 2016 | | 80.72 [ 77.44, 84.00] | 0.000 |
| Moyo et al., 2020 | | 80.76 [ 77.48, 84.05] | 0.000 |
| Moyo et al., 2020 | | 80.89 [ 77.59, 84.18] | 0.000 |
| Landes et al., 2021 | | 80.73 [ 77.45, 84.01] | 0.000 |
| Chagomerana et al., 2018 | | 80.80 [ 77.51, 84.08] | 0.000 |
| Landes et al., 2019 | | 80.72 [ 77.44, 84.00] | 0.000 |
| Chagomerana MB et al., 2021 | | 80.75 [ 77.46, 84.03] | 0.000 |
| Chagomerana MB et al., 2022 | | 80.71 [ 77.44, 83.99] | 0.000 |
| Chagomerana MB et al., 2018 | | 80.64 [ 77.37, 83.90] | 0.000 |
| Napyo et al., 2020 | | 80.65 [ 77.38, 83.91] | 0.000 |
| Munyaneza et al., 2025 | | 80.65 [ 77.38, 83.91] | 0.000 |
| M. Gill et al., 2016 | | 80.78 [ 77.50, 84.07] | 0.000 |
| Koss et al., 2017 | | 80.87 [ 77.58, 84.16] | 0.000 |
| Gabagaya et al., 2021 | | 81.83 [ 79.14, 84.51] | 0.000 |
| M. Humphrey et al., 2022 | | 80.73 [ 77.45, 84.00] | 0.000 |
| M. Humphrey et al., 2023 | | 80.63 [ 77.37, 83.89] | 0.000 |
| Sandbulte et al., 2020 | | 81.02 [ 77.75, 84.29] | 0.000 |
| Boisson-Walsh et al et al., 2024 | | 81.13 [ 77.88, 84.37] | 0.000 |
| Yotebieng et al., 2019 | | 81.21 [ 78.00, 84.42] | 0.000 |
| Yotebieng et al., 2020 | | 81.31 [ 78.14, 84.47] | 0.000 |
| Yotebieng et al., 2020 | | 81.15 [ 77.92, 84.39] | 0.000 |
| Woldesemayat et al., 2024 | | 80.54 [ 77.31, 83.77] | 0.000 |
| Hailu et al., 2025 | | 80.55 [ 77.32, 83.79] | 0.000 |
| Demissie et al., 2020 | | 80.69 [ 77.42, 83.97] | 0.000 |
| Alamneh et al., 2023 | | 80.67 [ 77.40, 83.94] | 0.000 |
| Aduloju et al., 2020 | | 81.04 [ 77.78, 84.31] | 0.000 |
| Musanhu et al., 2022 | | 80.73 [ 77.45, 84.01] | 0.000 |
| Duri et al., 2020 | | 80.91 [ 77.62, 84.19] | 0.000 |
| N. Atanga et al., 2018 | | 80.64 [ 77.37, 83.90] | 0.000 |
| Kafack et al., 2022 | | 80.70 [ 77.42, 83.97] | 0.000 |
| Ndlangamandla et al., 2023 | | 80.84 [ 77.55, 84.13] | 0.000 |
| M. Flynn et al., 2021 | | 80.96 [ 77.68, 84.25] | 0.000 |
| Mazuguni et al., 2021 | | 80.92 [ 77.64, 84.21] | 0.000 |
| Schrubbe et al., 2021 | | 81.18 [ 77.95, 84.40] | 0.000 |
| Schrubbe et al., 2022 | | 81.24 [ 78.04, 84.44] | 0.000 |
| Schrubbe et al., 2023 | | 81.13 [ 77.89, 84.38] | 0.000 |

Random-effects REML model

**Fig 4. A leave-one-out sensitivity analysis of virologic suppression among HIV-positive pregnant and lactating women on antiretroviral therapy in Africa.**

The odds of having virologic suppression were higher among women living in urban compared with rural resident women [64,95,97]. Furthermore, women's educational status was significantly associated with VS. Evidence from different studies consistently showed that women with secondary school or above had higher VS compared to women with no formal education [64,73].

**Table 2. Meta-regression for factors correlated with virologic suppression among HIV-positive pregnant and lactating women receiving antiretroviral therapy in Africa.**

| Variables | β-coefficient | Std. error | z | P-value | 95% CI |
|---|---|---|---|---|---|
| Publication year | | | | | |
| After 2020 (Ref.) | | | | | |
| Before 2020 | −2.73 | 3.55 | −0.77 | 0.443 | −9.69, 4.24 |
| ART regimen | | | | | |
| Non-DTG based (Ref.) | | | | | |
| DTG-based | 16.43 | 6.55 | 2.51 | 0.012 | 3.58, 29.26 |
| Country | −0.24 | 0.183 | −1.33 | 0.183 | −0.60, 0.12 |
| Region of Africa | −2.92 | 1.40 | −2.08 | 0.037 | −5.66, −0.17 |
| Sample size | −0.001 | 0.0002 | −1.30 | 0.195 | −0.0005, 0.0001 |
| Constant | 91.48 | 5.77 | 15.85 | 0.000 | 80.17, 102.80 |

Virologic suppression was not achieved among women who started ANC booking or initiated ART in the second or third trimester [44,64,95,98]. Similarly, long distance to reach a health facility, shortage of health providers, inadequate counseling, ART drugs running out, and sub-optimal sample transportation are linked with failure of virologic suppression [12,99,100]. In addition, advanced WHO stage of HIV (clinical stage II, III or IV) [39,98]; fear of stigma and discrimination [12,99,100]; and disclosure of HIV status [16,39,95,101,102] were found to be significantly associated with virologic suppression among HIV-positive pregnant and lactating women receiving ART (**S5 Table**).

In this meta-analysis, younger women (15–24 years) (AOR = 0.49; 95% CI: 0.32–0.77) were 51% less likely to attain virologic suppression compared to those aged 25 years and older [16,64,96]. Furthermore, a pooled analysis of three studies found that women who disclosed their HIV status (AOR = 1.66; 95% CI: 1.31, 2.11) were 1.66 times more likely to achieve virologic suppression compared to their counterparts.

In addition, the likelihood of achieving viral load suppression was 6.5 times higher for pregnant and lactating women on a first-line ART regimen (AOR = 6.53; 95% CI: 1.93, 22.06) than for those on second- or third-line regimens. Finally, the odds of virologic suppression among pregnant and lactating women who had good ART adherence (AOR = 3.61, 95% CI: 1.18, 11.02) were 3.61 times more likely compared to those who had poor adherence to ART (**Table 3**).

**Table 3. Meta-analysis for factors associated virologic suppression among HIV-positive pregnant and lactating women on antiretroviral therapy in Africa.**

| Variables | No_ studies | AOR (95% CI) | I² (%) | P-value |
|---|---|---|---|---|
| Maternal age | | | | |
| ≥25 years (Ref.) | 4 | | | |
| 15-24 years | | 0.49 (0.32, 0.77) | 79.7% | 0.002 |
| Disclosing HIV status to partner | 4 | 1.66 (1.31, 2.11) | 0.00% | 0.04 |
| ART type | | | | |
| First line ART | 3 | 6.53 (1.93, 22.03) | 86.07% | 0.001 |
| Second and/ third line ART (Ref.) | | | | |
| Adherence status to ART | 5 | | | |
| Good | | 3.61 (1.18, 11.02) | 94.6% | 0.001 |
| Poor (Ref.) | | | | |

## Discussion

The main objectives of this systematic review and meta-analysis were to determine the proportion of virologic suppression among pregnant and lactating women receiving ART and identify factors associated with it. Thus, the pooled prevalence of virologic suppression among pregnant and lactating women taking ART in Africa was 80.86% (95% CI: 77.63%, 84.09%). Additionally, the pooled estimate showed that only 60.92% of participants achieved an undetectable viral load (95% CI: 52.46%, 69.39%; $I^2 = 99.91\%$). This conclusion was comparable with a study conducted in East Africa, which found 80.6% virologic suppression among patients living with HIV. However, the finding of this review was lower than studies reported: 84% in Ethiopia [103], 85% in Sub-Saharan Africa [104], and far lower than the global target of achieving 95% viral suppression in patients receiving ART [31]. In contrast, the findings of the current study were higher than a study conducted in Ethiopia (71%) [105].

This could be due to the difference in the study population, study setting, and availability and adherence to ART. Thus, the study population for the previous studies was adults living with HIV, whereas the current study involves pregnant and lactating women receiving ART in Africa. The possible explanation could relate to physiological and immunological changes during perinatal periods [106]. Pregnancy alters the immune system and pharmacokinetics of ART drugs, which reduce the effectiveness of ART and increase maternal viral load [106,107].

Furthermore, delayed initiation or inadequate adherence to ART, perceived shame, and socioeconomic or health system constraints, such as limited regular VL surveillance, ART stock-outs, and a shortage of maternity professionals, could be the possible reason for non-virological suppression [16,59,108]. Suboptimal viral suppression or virologic failure jeopardizes not only the health of women but also raises the risk of HIV MTCT during pregnancy, childbirth, and breastfeeding, posing a challenge to the global effort towards eliminating pediatric HIV infection [59,109,110].

The UNAIDS's 95-95-95 goals for all groups of populations, including pregnant and lactating women, demonstrate the globe's ongoing dedication to combating HIV/AIDS and reaching the 2030 sustainable development goal of "ending AIDS" as a global health concern [32]. Despite considerable global progress towards reaching the 95-95-95 goals in the past decade, many low-middle-income countries, including many African countries, are still far from this goal [111]. Reaching UNAIDS's third 95% target, ensuring that 95% of people living with HIV on ART maintain sustained VS, is vital for preventing HIV transmission, improving health outcomes, and optimizing the quality of life [47]. Nonetheless, ART availability and adherence, ongoing structural and socioeconomic inequities, health-care system fragilities, and financing constraints Despite the hurdles, targeted interventions, including better ART adherence support, integrated maternal-HIV care, robust VL monitoring, and addressing structural, financial, and sociocultural barriers to close the gap and achieve the global target, ultimately lower vertical transmission and improve maternal and child health outcomes [47,112].

According to the findings of this study, Africa has not yet achieved the global target of attaining virological suppression in 95% of patients who received ART. To increase the possibility of successful virological outcomes, a variety of factors and problems must be addressed. In this systematic review, individual-level variables, healthcare utilization (time of ANC booking, time of ART initiation, being on a first-line ART regimen, duration of ART, baseline VL threshold, adherence status to ART), and barriers to healthcare services, including distance to health facility, shortage of health professionals, disclosure of HIV status, inadequate counseling, ART drug stock-out, and sub-optimal sample transportation, were significantly associated with virologic suppression among HIV-positive pregnant and lactating women in Africa.

In this study, the odds of virologic suppression among young women, particularly those 15–24 years old, were decreased by 51% when compared with those aged 25 years and above. The finding of this study was consistent with other studies [108,113–115]. Younger people receiving treatment for HIV might have difficulty with compliance due to various factors, including employment status, societal expectations, worries regarding stigma and disclosure, mental health challenges, and lacking health literacy [108,116]. In Africa, the stigma that exists at the individual, interpersonal, social, and organizational levels remain to pose significant hurdles for receiving and adhering to ART for adolescents living with HIV [117,118].

Antiretroviral therapy (ART) has changed the treatment of HIV from a deadly illness to a chronic condition that can be managed owing to its increased accessibility and continuing adherence [119]. Nevertheless, there are several obstacles in the adolescents HIV care cohort, which frequently put this age group at risk for less favorable outcomes than adults in the HIV care pathway. These issues include poorer adherence, lower retention in therapy, a lower rate of virological suppression, and greater rates of mortality [116,120–122].

Collaborated global, regional, and national efforts are needed to reach the UNAIDS 95% sustained viral suppression rate, especially for adolescents living with HIV, who are frequently linked with elevated rates of virological failure (VF) [32,112]. Understanding and addressing the unique challenges faced by young women helps healthcare workers and decision-makers create specific treatments that boost viral suppression rates and enhance health for this vulnerable age group. Therefore, promoting ART adherence support, regular VL monitoring, and accessible and flexible adolescent-youth-friendly HIV care, while guaranteeing accessibility, privacy, and a favorable environment, plays a pivotal role in achieving VS in these groups of populations [117,118].

The odds of achieving virologic suppression were higher among women living in urban areas than women residing in rural areas. This finding was consistent with previous study findings [64,95,97]. This could be due to women living in urban areas being more likely to have better awareness and access to HIV care, including periodic VL monitoring and continuous ART supply, as well as enhanced counseling on ART adherence. Whereas rural women frequently encounter geographical, infrastructural, and service-related barriers that impede optimal ART adherence and VS [95].

Furthermore, women's educational status was significantly associated with virologic suppression, as studies consistently show that women with secondary or higher education are more likely to achieve VS than those with no formal education [64,73]. This could be explained by the fact that education increases awareness of navigating medical care, the importance of ART adherence, and access to HIV care; in contrast, less educated women could find it challenging to grasp treatment, receive regular medical care, and fear stigma. Moreover, pregnant and breastfeeding women who disclosed their HIV status were significantly associated with achieving virological suppression compared to those who did not. This finding is in agreement with other studies that revealed disclosure of HIV status increases the likelihood of virologic suppression [16,36,39,95,101,102,103,108,123]. Disclosing one's HIV status to family, friends, and intimate partners can have positive health impacts. Several studies show that individuals who disclosed their serostatus had better social assistance, stronger family and relationship solidarity, lower symptoms of depression and anxiety, improved physical health, psychological assistance, financial support, and better adherence to ART [124–126]. However, fear of stigma and discrimination was found to be significantly associated with virologic suppression among HIV-positive pregnant and lactating women receiving ART [12,99,100,108]. Stigma can take many forms, including societal discrimination, internalized prejudice, family stigma, and stigma in healthcare environments. This may result in fear of disclosing status or withholding drugs, lack of self-worth, and failure to adhere to ART [45,127,128].

Women who began ANC contacts or started ART in the second or third trimester were less likely to achieve virologic suppression [44,64,95,98]. A late beginning of ANC follow-up and ART decreases the time period of effective ART for decreasing VL before childbirth and limits chances for regular monitoring of ART treatment outcomes, enhanced counseling for adherence, and prompt response to treatment challenges, which are essential for attaining and sustaining optimum virologic suppression to prevent MTCT of HIV.

Advanced WHO HIV stages (II, III, or IV), marked by opportunistic infections, were significantly associated with failure to achieve virologic suppression among pregnant and lactating women on ART [39,98,108]. Advanced HIV stages elevate the likelihood of virologic suppression failure because growing immune suppression and opportunistic infections may hamper ART efficacy, alter adherence, and increase the risk of treatment failure, thereby rendering VL control more challenging for pregnant and breastfeeding women [39,108].

This study also found that patients on first-line ART regimens had higher rates of virologic suppression compared to those on second- or third-line regimens, consistent with findings from other studies [36,37,129,130]. This could be

explained as first-line ART drugs (TDF-3TC-DTG) having higher efficacy (sustained VL suppression), fewer adverse effects, and lower resistance when compared to second- and third-line ART regimens [25,131,132]. Therefore, all HIV-positive pregnant and lactating women should initiate triple ART as early as possible, regardless of their CD4 count or WHO clinical stage. This leads to optimal suppression of viral load to undetectable levels, thereby preventing the vertical transmission of HIV [133]. Nevertheless, the virus can never be totally eliminated from the human body, so an individual should continue to take the treatments even after the signs and symptoms have subsided and adhere to safer sexual practices [25,131,134].Furthermore, good adherence to antiretroviral therapy was significantly associated with virologic suppression among pregnant and breastfeeding women, aligning with findings from previous studies [36,37,39,129]. The plausible explanation is due to the fact that good adherence to ART sustains maximal viral load suppression to undetectable levels, lowers drug resistance, and boosts the immune system (CD4 cell counts), which leads to substantially lowering the risk of mother-to-child transmission of HIV and improving the overall health of individuals [131,135]. However, adherence to ART remains a major health issue worldwide, particularly in low- and middle-income nations. Virologic and clinical outcomes are highly dependent on strong compliance to ART, while poor adherence to antiretroviral drugs has been found to be a significant predictor of virologic failure, drug resistance, progression of disease, admissions to hospitals, high death rate, and medical care expenses [108,136,137].

On the other hand, long distance to reach a health facility, shortage of health providers, inadequate counseling, ART drugs running out, and shortage of HIV commodities linked are with failure of virologic suppression [12,45,95,99,100]. [45,138]. Health facility availability is a crucial infrastructural variable determining adherence to ART, especially in low-income countries, in which medical infrastructure is frequently poor. Individuals with work or domestic responsibilities have significant challenges due to the facility's limited operation hours or inconvenient hours, which are often only available during weekday daytime [45].

Similarly, ART stockouts in healthcare facilities result in viral rebounding, missed ART doses, drug resistance, and switching to the second-line therapy [45,139–141]. ART drugs can occasionally be inaccessible due to supply chain issues or financing shortages in several developing countries. Patients may receive an insufficient supply or nothing at all as they come for refills following a stockout, which would cause them to skip medications [45,141,142].

In general, the findings of this review have various implications for clinical practice, policy, and future research works in these population groups. Since lifelong first-line ART regimens and enhanced adherence to ART have been strongly linked to greater virologic suppression, healthcare practitioners should increase adherence counseling, particularly during prenatal and postpartum contacts. The lower suppression rates in younger women demonstrate the necessity for youth-specific strategies like youth-friendly clinics, psychosocial assistance, and sexual and reproductive health education incorporated into HIV treatment programs. Optimizing virologic suppression in pregnant and breastfeeding women on ART in Africa requires a comprehensive strategy that includes upgrading clinical practices, ensuring responsive policy, and ongoing VL monitoring and evaluation to address ongoing barriers, gaps, and disparities to achieve VS. Further qualitative and quantitative studies are required to investigate different sociocultural, behavioral, psychological, and structural factors that lead to virologic non-suppression among HIV-positive pregnant and breastfeeding women in Africa.

### Strengths and limitations of the study

To our knowledge, this would be the first systematic review and meta-analysis to assess the overall estimate of virologic suppression and associated factors among HIV-positive pregnant and breastfeeding women in Africa. We have searched multiple databases to include both published and unpublished studies and followed a meticulous screening approach to compile all eligible studies. Thus, the findings of this review provide important baseline evidence for health providers, policymakers, academicians, and global partners to design and implement evidence-based interventions to achieve the WHO 95-95-95 global target to eliminate vertical transmission of HIV.

However, the following limitations should be considered when interpreting the findings of this review. Despite the fact that enough primary studies were used to estimate the overall viral load suppression, many countries and some African regions were not evenly represented. The presence of significant heterogeneity among the included studies might affect the generalizability of the finding. This might be due to variations in the study area and the methodological quality of the studies, such as study design, sampling technique, response rate, sample size, and study setting, that could alter the pooled estimate's generalizability. Additionally, due to the concentration of studies in some countries, such as South Africa, it could restrict the conclusion of the finding to the entire continent. To account this variability, we employed a random-effects model for analysis in order to overcome this significant variability. Additionally, we performed subgroup analyses, sensitivity analyses, and meta-regression to identify potential causes of substantial between-study heterogeneity.

Future studies should focus on longitudinal and mixed-methods designs that are geographically representative and methodologically robust to gain deeper understanding of virologic suppression during pregnancy, childbirth, and the postpartum phase. Longitudinal studies can show the temporal relationship between viral load and associated factors; whereas mixed-methods approaches can deepen understanding of the sociocultural, behavioral, and structural facilitators and barriers influencing virologic suppression in these targeted populations. In order to guide focused interventions and promote the elimination of perinatal transmission of HIV in various African contexts, it is imperative to generate such robust and scientifically grounded evidence.

## Conclusions

According to this systematic review and meta-analysis, the overall prevalence of virologic suppression among pregnant and lactating women taking ART in Africa (80.86%) is lower than USAID's 95% global target of VL suppression. In this study, socio-demographic characteristics (women's age, higher educational level, being married/cohabitant, and urban residency) and healthcare utilization (time of ANC booking, time of ART initiation, being on a first-line ART regimen, duration of ART, and adherence status to ART) were significantly associated with virologic suppression among HIV-positive pregnant and lactating women in Africa.

Additionally, virologic suppression is also substantially linked with various facilitators and barriers, including disclosing of HIV status, fear of stigma, distance to health facility, shortage of health professionals, ART drug stock-out, and lack of HIV care commodities. This highlights the importance of targeted interventions for young HIV-positive women, encouraging disclosing HIV status, initiating first-line antiretroviral therapy regimens, and promoting antiretroviral therapy adherence; strengthening healthcare systems that offer regular viral load monitoring; and addressing socio-demographic and ART-related factors as vital steps toward improving treatment outcomes and reducing perinatal transmission of HIV.

## Supporting information

**S1 File. PRISMA 2020 checklist.**
(DOCX)

**S2 File. Search strategies for virologic suppression among HIV-positive pregnant and lactating women receiving ART in Africa.**
(DOCX)

**S3 File. Excel data extraction spreadsheet for virologic suppression among HIV-positive pregnant and lactating women in Africa.**
(XLSX)

**S4 Table. Descriptive summary of studies included in systematic review of virologic suppression among HIV-positive pregnant and lactating women on ART in Africa.**
(DOCX)

**S5 Table. Factors associated with virologic suppression among HIV-positive pregnant and lactating women receiving ART in Africa.**
(DOCX)

## Acknowledgments

The authors of this study would like to express our heartfelt gratitude to all of the authors of the studies included in this systematic review and meta-analysis.

## Author contributions

**Conceptualization:** Berihun Agegn Mengistie, Gebrye Gizaw Mulatu, Nuhamin Tesfa Tsega.

**Data curation:** Berihun Agegn Mengistie, Kindu Yinges Wondie, Alemken Eyayu Abuhay, Tazeb Alemu Anteneh, Habtu Kifle Negash, Nuhamin Tesfa Tsega.

**Formal analysis:** Berihun Agegn Mengistie, Getie Mihret Aragaw, Kindu Yinges Wondie, Alemneh Tadesse Kassie, Alemken Eyayu Abuhay, Wondimnew Mersha Biset, Abay Eyayu Asrie, Habtu Kifle Negash, Nuhamin Tesfa Tsega.

**Methodology:** Berihun Agegn Mengistie, Getie Mihret Aragaw, Gebrye Gizaw Mulatu, Alemken Eyayu Abuhay, Moges Tesfa Tsega, Eshet Gebrie, Nuhamin Tesfa Tsega.

**Software:** Berihun Agegn Mengistie, Getie Mihret Aragaw, Alemneh Tadesse Kassie, Alemken Eyayu Abuhay, Wondimnew Mersha Biset, Abay Eyayu Asrie.

**Supervision:** Alemken Eyayu Abuhay.

**Validation:** Berihun Agegn Mengistie, Getie Mihret Aragaw, Kindu Yinges Wondie, Eshet Gebrie, Nuhamin Tesfa Tsega.

**Visualization:** Gebrye Gizaw Mulatu, Alemken Eyayu Abuhay, Abay Eyayu Asrie.

**Writing – original draft:** Kindu Yinges Wondie, Alemneh Tadesse Kassie, Alemken Eyayu Abuhay, Wondimnew Mersha Biset, Moges Tesfa Tsega, Abay Eyayu Asrie, Tazeb Alemu Anteneh, Habtu Kifle Negash, Nuhamin Tesfa Tsega.

**Writing – review & editing:** Berihun Agegn Mengistie, Getie Mihret Aragaw, Gebrye Gizaw Mulatu, Kindu Yinges Wondie, Alemken Eyayu Abuhay, Eshet Gebrie, Nuhamin Tesfa Tsega.

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
