## [Decision Letter · Decision Letter 0]

27 Oct 2025

Dear Dr. Mengistie,

Thank you for submitting your manuscript to PLOS ONE. After careful consideration, we feel that it has merit but does not fully meet PLOS ONE’s publication criteria as it currently stands. Therefore, we invite you to submit a revised version of the manuscript that addresses the points raised during the review process.

We look forward to receiving your revised manuscript.

Kind regards,

Richard Makurumidze (MBChB, MPH, PhD, FCPHP)

Academic Editor

PLOS ONE

Journal Requirements:

2. Please include a separate caption for each figure in your manuscript.

Reviewers' comments:

Reviewer's Responses to Questions

**Comments to the Author**

1. Is the manuscript technically sound, and do the data support the conclusions?

Reviewer #1: Partly

Reviewer #2: Yes

Reviewer #3: Yes

Reviewer #4: Partly

Reviewer #5: Yes

Reviewer #6: Yes

Reviewer #7: Yes

Reviewer #8: Yes

Reviewer #9: Yes

Reviewer #10: Yes

Reviewer #11: Yes

Reviewer #12: Yes

2. Has the statistical analysis been performed appropriately and rigorously?

Reviewer #1: No

Reviewer #2: Yes

Reviewer #3: Yes

Reviewer #4: No

Reviewer #5: Yes

Reviewer #6: Yes

Reviewer #7: Yes

Reviewer #8: Yes

Reviewer #9: I Don't Know

Reviewer #10: No

Reviewer #11: Yes

Reviewer #12: Yes

3. Have the authors made all data underlying the findings in their manuscript fully available?

Reviewer #1: Yes

Reviewer #2: Yes

Reviewer #3: Yes

Reviewer #4: Yes

Reviewer #5: Yes

Reviewer #6: Yes

Reviewer #7: Yes

Reviewer #8: Yes

Reviewer #9: Yes

Reviewer #10: Yes

Reviewer #11: Yes

Reviewer #12: Yes

4. Is the manuscript presented in an intelligible fashion and written in standard English?

Reviewer #1: Yes

Reviewer #2: Yes

Reviewer #3: Yes

Reviewer #4: Yes

Reviewer #5: Yes

Reviewer #6: Yes

Reviewer #7: Yes

Reviewer #8: Yes

Reviewer #9: Yes

Reviewer #10: No

Reviewer #11: Yes

Reviewer #12: Yes

Reviewer #1: The manuscript addresses a critical public health concern by evaluating virologic suppression among HIV-positive pregnant and lactating women in Africa. Its significance is evident given the implications for mother-to-child transmission and the broader context of achieving global HIV targets. The inclusion of 55 studies and over 300,000 participants strengthens the analysis through a large sample size, and the use of PRISMA guidelines, PROSPERO registration, and appropriate statistical tools such as random-effects models and Egger’s tests demonstrate methodological rigor. The study presents relevant findings with public health utility, highlighting key factors such as age, adherence, disclosure of HIV status, and ART regimen as significantly associated with viral suppression.

Despite these strengths, several issues warrant attention. A major concern lies in the extremely high heterogeneity (I² > 99%), which remains unexplained even after subgroup analyses. The use of meta-regression would have been appropriate to explore potential sources of heterogeneity more systematically. The overrepresentation of studies from South Africa introduces geographical bias, making it difficult to generalize findings across the diverse healthcare contexts of the African continent. Furthermore, although the authors state there was no publication bias, the Egger’s test result (p = 0.0512) suggests borderline significance, which should not be dismissed as negligible without further investigation.

The study also overlooks key confounding factors such as socioeconomic status, healthcare accessibility, or rural–urban disparities, which are known to influence ART outcomes. In addition, the reporting lacks clarity in some areas. The year range for included studies curiously extends to 2025, which undermines credibility and suggests a typographical or planning oversight. The quality assessment using the Joanna Briggs Institute tool is not detailed, and no summary of individual study scores is provided, which limits transparency. Wide confidence intervals for certain effect estimates (e.g., ART adherence) further point to imprecision or potential small-study effects.

Narratively, the discussion is repetitive, often restating results without providing deeper contextual interpretation or drawing clear distinctions between policy implications and existing literature. The paper also misses the opportunity to provide targeted, practical recommendations for policymakers, especially considering its relevance to achieving UNAIDS 95-95-95 targets. Finally, factors such as ART duration, viral load assay variability, and breastfeeding practices, which could significantly influence virologic suppression, are insufficiently addressed. Overall, while the study presents valuable findings, it would benefit from a more nuanced analytical approach, clearer methodological transparency, and a stronger focus on contextual and actionable insights.

Reviewer #2: thank you for your interested topic

The study is scientifically relevant, methodologically solid, and has potential impact for HIV programmatic policy in Africa.

I advice to Revise for minor grammar mistakes, consistent abbreviation use, and concise phrasing.

Reviewer #3: 1. Does the title accurately reflect the study’s objectives and population?

2. The abstract mentions an 80.92% pooled prevalence should confidence intervals or heterogeneity measures (I2) also appear for completeness?

3. Were the inclusion and exclusion criteria too broad (2016–2025) given the variability in ART regimens over time?

4. How were overlapping datasets or duplicate reports from large cohort programs handled?

5. Were data extraction and bias assessments blinded and independently verified by more than two reviewers?

6. How did the authors deal with variations in the definition of “virologic suppression” across studies (e.g., ≤50 vs ≤1000 copies/mL)?

7. Given the high heterogeneity (I2 = 99.57%), was the use of a pooled random-effects model justified without meta-regression?

8. Could differences in study design (cross-sectional vs. cohort) have contributed to heterogeneity, and were these tested?

9. How were missing data handled in the included studies?

10. Could differences in viral-load testing availability or assay methods influence reported suppression rates?

I appreciate the authors efforts of gathering huge data and preparing. I am making a request to author to answer these question which are necessary.

Reviewer #4: Thank you for the opportunity to review this manuscript.

The topic, “virologic suppression among HIV-positive pregnant and lactating women in Africa” is important and policy-relevant.

With a few targeted methodological clarifications and stronger transparency, this can become a solid, reproducible synthesis.

Major comments

1) Outcome definition & measurement window

Please harmonize the outcome definition and clarify, at study level, the VL threshold used (e.g., ≤1000 vs other), specimen type (plasma vs DBS), and timing (pregnancy trimester / intrapartum / postpartum window). Provide either:

• a succinct subgroup/meta-regression using these moderators; or

• a sensitivity analysis restricted to ≤1000 copies/mL.

This avoids definitional mixing and improves interpretability.

2) Modeling proportions under extreme heterogeneity

Given the very high between-study heterogeneity (I² ≈ 99.6%), please complement DerSimonian–Laird with HKSJ or REML, and add a logit/GLMM approach suitable for proportions (with appropriate continuity correction where needed).

In the main text, report τ² and a 95% prediction interval (PI) alongside the pooled mean.

3) Explaining heterogeneity rather than averaging it away

Your region/country subgrouping is helpful. If feasible, consider a compact set of clinically/programmatically meaningful moderators, ART era (EFV vs DTG rollout) and maternal age (15-24 vs ≥25 years), via simple stratified analyses or a brief meta-regression.

If further analyses are not practical, a short justification for the current scope plus a more cautious interpretation (with the PI) is acceptable.

4) Small-study effects / publication bias

Under heavy heterogeneity, Egger is suboptimal for single-arm proportions. Please consider Peters/Harbord tests where the number of studies permits and temper funnel-plot interpretations accordingly; if underpowered, state this explicitly.

5) Unit of analysis & potential double counting

Where a single dataset contributes multiple effects (e.g., pregnant vs intrapartum vs postpartum; or multiple VL thresholds), please explain how you avoided double counting, for example, by pre-specifying one time-point, aggregating within-study effects, or using a multilevel model.

6) Risk of bias & transparency (search, RoB, data/code)

Include a per-study JBI risk-of-bias matrix in the supplement, justify any scoring thresholds, and add a sensitivity analysis excluding higher-risk studies. For search reproducibility, provide full search strings, last search date, and the deduplication procedure; also clarify whether Embase/Scopus and Africa-focused sources (e.g., AIM/AJOL) were searched or provide a documented rationale for their exclusion.

For reproducibility, make the extraction sheet (study-level events/total and moderators) fully available per PLOS policy (as SI or in a DOI-minted repository). Sharing analysis code (Stata/R) is strongly encouraged; at minimum, include sufficient model/command detail for re-analysis.

7) Determinants / associated factors (AORs)

Please specify the eligibility criteria for studies entering the determinants analysis and ensure alignment of reference categories and reasonable consistency of adjustment sets before pooling on the log-OR scale. If adjustment sets differ substantially or heterogeneity is high with wide CIs, consider a narrative synthesis or meta-regression rather than a pooled AOR. Clarify why only a subset of studies contributed and whether others were excluded for design/measure reasons.

Minor comments

• Move very long descriptive tables to the supplement and keep concise summary tables in the main text.

• Include the 95% prediction interval in the Abstract/Results so readers immediately see expected variability across settings.

• Define acronyms at first use (VS/VLS, PMTCT, DTG/EFV) and streamline a few long sentences with a light language edit.

• Proofread tables for numerical slips (e.g., any CI exceeding 100%) and ensure consistent terminology.

• Generalizability: where certain sub-analyses rely heavily on national program data, please comment on applicability to smaller or non-program settings.

• Briefly cite the PROSPERO registration (ID) and ensure the PRISMA flow diagram is referenced in the main text for reproducibility.

Recommendation: Major Revision.

The manuscript is valuable; the requested clarifications, outcome harmonization, robust proportion modeling, focused heterogeneity exploration (including ART era and age), and transparent materials (search/RoB/extraction file and, ideally, code), will make the conclusions more reliable and reproducible.

Reviewer #5: This manuscript presents a systematic review and meta-analysis assessing the pooled prevalence of virologic suppression (VS) among HIV-positive pregnant and lactating women receiving antiretroviral therapy (ART) in Africa. The authors included 55 studies encompassing 304,883 participants across various African regions. The study followed PRISMA guidelines and used robust statistical methods, including a random-effects model, subgroup analyses, and heterogeneity assessment.

The topic is highly relevant, particularly in the context of achieving the global UNAIDS 95–95–95 targets and improving maternal and child health outcomes in sub-Saharan Africa. The manuscript contributes meaningful evidence regarding ART adherence and determinants of viral suppression in this key population group. This manuscript has major strengths such as relevance, scope and data size, systematic approach, robust analysis, policy value.

Minor Editorial Comments:

In Abstract section , replace “remains low” with quantitative context (“approximately 81%, below global target of 95%”). Conduct meta-regression and sensitivity analysis by study design, ART regimen, and publication year.

Provide more discussion on possible small-study or regional publication bias.

Include qualitative evidence or secondary data review to contextualize findings.

Ensure all forest plots (Figures 2–8) have consistent formatting and readable axis labels.

For References, several citations lack full journal details or DOI (e.g., refs. 70, 77). Please verify all references conform to PLOS ONE style. Conduct sensitivity analysis by publication period (pre/post-2020.

Minor edits needed for flow in Introduction and Discussion to grammar.

Closing Statement:

This manuscript is a well-executed, timely, and policy-relevant meta-analysis that adds significant value to the understanding of virologic suppression among HIV-positive women in Africa.

With minor clarifications on methodology, expanded heterogeneity analysis, and slight editorial adjustments, the paper will meet PLOS ONE’s publication standards and provide actionable insights for maternal HIV prevention and control programs across the continent.

Reviewer #6: Authors section: I suggest you only use Names, instead of Initials

Delete, Let’s only have correspondent Authors. The email addresses for each will be put in the PLOS ONE form as you submit the manuscript

Add Problem Statement, and Objectives. Research questions without Problem statement is hanging. I suggest you add

1. Problem statement

2. Significancy of this Systematic Review

Refer to the attached manuscript with tracj changes

Reviewer #7: Page 139-143 : The problem needs to be further refined, especially for research issues concerning viral load viruses. Ethical clearance issues also sometimes become obstacles and the lack of study in HIV research

......

page 168 -175 : Please clarify the source of the data taken, because in the background of the problem, you describe something that highlights the lack of research in the field regarding this topic.

.......

page 194 -200 : Could you provide more details on the research method conducted with a pooled prevalence, including which data was used and the operational definitions applied?

....

page 211- 220 : is it only using this data to determine the pooled prevalence ?

....

page 231-233 : Why you leave-one-out sensitivity anlysis ?

...

page 255 - 256 : can you more explain with this result ?

...

page 351 -356 : This explanation must be further clarified and supported by literature that underpins this statement.

...

page 417 : Although this research is a systematic review and meta-analysis using secondary data, researchers must still consider intellectual property rights and proper attribution for the data used.

Reviewer #8: Given the high prevalence of HIV-positive women in African countries, it is important to understand virological suppression in pregnant and breastfeeding women. Conducting this systematic review provides insight into the current status of this population.

The authors have considered the necessary methodological criteria and clearly described the process for developing this systematic review. However, it is not entirely clear how they analyzed and interpreted the determinants of virological suppression and associated variables, nor how they addressed potential confounders. It would be helpful to detail these aspects in the review manuscript.

When evaluating individual studies, why was the Critical Appraisal Checklist considered instead of a validated tool for assessing the methodological quality of studies? Potential biases arising from the study assessment are not addressed in the limitations section of the review. Please comment.

Reviewer #9: Thank you for doing research on his interesting topic. It is essential to emphasize the importance of implementing targeted interventions for young HIV-positive women, including the disclosure of HIV status, initiation of first-line antiretroviral therapy (ART) regimens, and promotion of ART adherence.

Reviewer #10: Review comments on “Virologic suppression among HIV-positive pregnant and lactating women receiving antiretroviral therapy in Africa: A systematic review and meta-analysis (PONE-D-25-49195)”

The manuscript (PONE-D-25-49195) addresses a critical public health issue concerning HIV viral suppression among pregnant and lactating women on antiretroviral therapy (ART) in Africa. The topic aligns well with global health and HIV epidemiology. The study is comprehensive but requires significant improvements in methodological transparency, reporting, and interpretation. I have the following concerns which if addressed will improve the study.

1.Prospero has CRD420251077063, registered as “Knowledge of mother-to-child transmission of HIV and its prevention among reproductive-age women in Africa: a systematic and meta-analysis 2025”.

- Register your review again in Prospero and state the ID correctly.

2. Under introduction (line 144 – 153), the research question is a repetition from the goal of the study. Merge the goal and research question to avoid repetition.

3. Remove (line 168- 175) and present as part of your supplementary file S2 file.

4. Include a complete PRISMA 2020 checklist and flow diagram under the results as part of the manuscript not part of list of figures (Figure 1). Draw it nicely or get it the flow diagram rom online, present it well in the manuscript, with well labelled figure legend.

5. For the risk of bias assessment, present the JBI quality assessment results in a summary table or figure.

6. Table 1 should be added to the results section with a well described table legend under the table, stating what criteria or reference was used for the risk of bias.

7. Table 2 should be added to the results section, not as part of list of tables, then with a legend describing or stating what was used to achieve the sub-group analysis.

8. Figure 1-8 should be presented as part of the results with a well labelled figure legend, stating how it was achieved.

9. Consider sources of heterogeneity and use meta-regression by study year, region/country, ART regimen and viral load threshold to improve the robustness of the findings.

10.Include in the discussion, a more clearly link between the UNAIDS 95-95-95 targets, national PMTCT programs and strategies to address low suppression in younger women.

Overall, this review is timely and relevant. Addressing the above comments will substantially enhance its scientific rigor, methodological transparency, and readability.

END.

Reviewer #11: Overall, this is good research, this review takes a deep dive into how well HIV-positive pregnant and

breastfeeding women in Africa are responding to antiretroviral therapy, specifically looking at how

many are achieving virologic suppression. The study is methodologically solid, using validated tools

and statistical models to analyze the data. It also identifies key factors that influence outcomes, like

age, relationship status, ART regimen, HIV status disclosure, and adherence. Younger women and

those with poor adherence were less likely to achieve suppression, pointing to the need for youth-

friendly care and stronger support systems. While the results are promising, the high variability

across studies and regions suggests more targeted interventions and further research are needed to

close the gap and reduce mother-to-child transmission

Reviewer #12: 1. General Assessment

This manuscript presents a systematic review and meta-analysis evaluating the prevalence of virologic suppression and its associated factors among HIV-positive pregnant and lactating women receiving ART in Africa. The study is timely, methodologically sound, and addresses a critical gap in maternal HIV care across the continent. The authors adhered to PRISMA guidelines, registered their protocol on PROSPERO, and employed appropriate statistical techniques including random-effects modeling, subgroup analysis, and sensitivity testing. The manuscript is generally well-written and organized, with clear objectives and relevant findings. However, several areas require clarification or refinement to meet PLOS ONE’s standards for transparency, reproducibility, and scientific rigor.

2. Section-by-Section Evaluation

Methods and Materials

Areas for Improvement:

1. Search Strategy: The Boolean search string is extensive but lacks structure. Present full search strategies (e.g., for PubMed) in a supplementary file.

2. Database Justification: Justify inclusion of Google Scholar due to its non-standard indexing.

3. Quality Threshold: Rephrase “excellent quality” for studies scoring ≥5 on JBI to “acceptable quality.”

4. Outcome Definitions: Clarify whether all studies used the same VL threshold (≤1000 copies/mL).

5. Statistical Rationale: Justify use of DerSimonian and Laird’s method over alternatives like Hartung-Knapp.

6. Publication Bias Tests: Clarify whether Egger’s and Begg’s tests were applied to both prevalence and AORs.

Results

Areas for Improvement:

1. PRISMA Flow: Include a complete PRISMA diagram and a table of excluded studies with reasons.

2. Heterogeneity: I² = 99.57% is extremely high. Consider meta-regression or deeper exploration of sources.

3. Subgroup Analysis: Report statistical significance of subgroup differences and I² values per subgroup.

4. Associated Factors: Provide AORs and CIs for all discussed predictors. Clarify model type used for pooling.

5. Figures: Ensure all figures (e.g., funnel plot, forest plots) are clearly labeled and referenced.

Discussion, Strengths & Limitations, Conclusion

Areas for Improvement:

1. Interpretation: Discuss clinical significance of the 14% gap from the 95% target.

2. Causal Language: Soften causal claims to reflect observational nature (e.g., “associated with”).

3. Policy Framing: Reference WHO/UNAIDS frameworks when recommending interventions.

4. Limitations: Clarify geographic skew and discuss potential impact of publication bias despite non-significant tests.

5. Conclusion: Separate findings from recommendations for clarity. Specify which health system components need strengthening.

3. Minor Editorial Suggestions

• Correct grammar and phrasing (e.g., “All searching databases” → “All searched databases”).

• Ensure consistent terminology: “virologic suppression” vs. “viral load suppression.”

• Fix typos (e.g., “studies is required” → “studies are required”).

.

Reviewer #1: **Yes:** Salvatore ChirumboloSalvatore ChirumboloSalvatore ChirumboloSalvatore Chirumbolo

Reviewer #2: No

Reviewer #3: **Yes:** sri chakradhar challagalisri chakradhar challagalisri chakradhar challagalisri chakradhar challagali

Reviewer #4: No

Reviewer #5: No

Reviewer #6: **Yes:** James ArinaitweJames ArinaitweJames ArinaitweJames Arinaitwe

Reviewer #7: **Yes:** JANNO B B BERNADUSJANNO B B BERNADUSJANNO B B BERNADUSJANNO B B BERNADUS

Reviewer #8: No

Reviewer #9: No

Reviewer #10: No

Reviewer #11: **Yes:** Oluwatobi AtolagbeOluwatobi AtolagbeOluwatobi AtolagbeOluwatobi Atolagbe

Reviewer #12: **Yes:** Duressa Shafi AhmedDuressa Shafi AhmedDuressa Shafi AhmedDuressa Shafi Ahmed

---

## [Author Response · Author response to Decision Letter 1]

27 Dec 2025

Dear Editorial Team,

Thank you for timely feedback and carefully reviewing our manuscript, "Virologic suppression among HIV-positive pregnant and lactating women receiving antiretroviral therapy in Africa: A systematic review and meta-analysis" (PONE-D-25-49195), and bringing to our attention the concerns about text overlap with previous articles.

We sincerely apologize for being late to revise our submitted manuscript. We are grateful to the academic editor for facilitating our submission and to the reviewers for their valuable scientific comments, expertise, and guidance, which were instrumental in improving the overall quality of our manuscript. The authors diligently and meticulously address all the comments and recommendations provided by all reviewers and the academic editor. As a result, we believe that the manuscript substantially improved in terms of grammar, punctuation, coherence, clarity, methodological rigor, transparency, and reproducibility. We have submitted the revised manuscript alongside the manuscript with track changes, "Response to Reviewers," and other relevant supplementary files.

We look forward to your timely feedback for our revised manuscript.

Thank you for your time and ongoing support.

---

## [Decision Letter · Decision Letter 1]

4 Feb 2026

Dear Dr. Mengistie,

Thank you for submitting your manuscript to PLOS ONE. After careful consideration, we feel that it has merit but does not fully meet PLOS ONE’s publication criteria as it currently stands. Therefore, we invite you to submit a revised version of the manuscript that addresses the points raised during the review process.

We look forward to receiving your revised manuscript.

Kind regards,

Richard Makurumidze

Academic Editor

PLOS One

Journal Requirements:

Reviewers' comments:

Reviewer's Responses to Questions

**Comments to the Author**

Reviewer #1: (No Response)

Reviewer #2: All comments have been addressed

Reviewer #3: All comments have been addressed

Reviewer #6: All comments have been addressed

Reviewer #7: All comments have been addressed

Reviewer #9: All comments have been addressed

Reviewer #10: All comments have been addressed

2. Is the manuscript technically sound, and do the data support the conclusions?

Reviewer #1: Partly

Reviewer #2: Yes

Reviewer #3: Yes

Reviewer #6: (No Response)

Reviewer #7: Yes

Reviewer #9: Yes

Reviewer #10: Yes

3. Has the statistical analysis been performed appropriately and rigorously?

Reviewer #1: No

Reviewer #2: Yes

Reviewer #3: Yes

Reviewer #6: (No Response)

Reviewer #7: Yes

Reviewer #9: I Don't Know

Reviewer #10: Yes

4. Have the authors made all data underlying the findings in their manuscript fully available?

Reviewer #1: Yes

Reviewer #2: Yes

Reviewer #3: Yes

Reviewer #6: Yes

Reviewer #7: Yes

Reviewer #9: Yes

Reviewer #10: Yes

5. Is the manuscript presented in an intelligible fashion and written in standard English?

Reviewer #1: No

Reviewer #2: Yes

Reviewer #3: Yes

Reviewer #6: No

Reviewer #7: Yes

Reviewer #9: Yes

Reviewer #10: Yes

Reviewer #1: I did not find a point-by-point reply to my recommendations. Next time I would like to find them. I read the revised paper. The revised manuscript effectively addresses most of the reviewer’s concerns. It now includes a meta-regression analysis that identifies significant moderators of heterogeneity, acknowledges and discusses the geographic concentration of studies in Southern Africa, and presents expanded discussion on key contextual factors such as socioeconomic status, healthcare access, and rural–urban disparities. The authors have also improved the narrative quality of the discussion, providing clearer interpretations and actionable policy recommendations aligned with the UNAIDS 95-95-95 goals. Additionally, they incorporated relevant factors such as ART duration, viral load assay thresholds, and breastfeeding practices, which were previously under-addressed.

However, a few issues remain unresolved. The manuscript still refers to an implausible study inclusion period extending to April 2025, which undermines credibility. Moreover, while the Joanna Briggs Institute tool is used for quality assessment, the authors do not provide a detailed summary of individual study scores, limiting transparency. These remaining concerns should be addressed to fully satisfy the reviewer’s expectations.

Reviewer #2: The manuscript presents a strong and timely systematic review and meta-analysis addressing virologic suppression among HIV-positive pregnant and lactating women in Africa.

Reviewer #3: I would like to begin by commending the authors for undertaking such a comprehensive and high-impact study. Synthesizing data from over 304,000 participants across 55 studies provides an essential view of the PMTCT landscape in Africa. The focus on the specific window of pregnancy and lactation is vital, as this demographic is often underserved in broader adult HIV cohorts. I am appreciating the author for his interest.

Reviewer #6: I would like to thank the authors for the considerable effort invested in preparing this manuscript. The study is clearly structured, well-written, and presents its objectives, methodology, and findings in a coherent and logical manner. The topic addressed is timely and relevant, and the work makes a meaningful contribution to the existing body of literature in this field.

The authors demonstrate a solid understanding of the theoretical background and prior research, and the literature review is appropriately comprehensive and up to date. The methodology is sound and sufficiently detailed to allow for reproducibility, and the analysis is carried out rigorously. The results are clearly presented and thoughtfully interpreted, with conclusions that are well supported by the data.

I particularly appreciate the clarity of the discussion section, where the authors effectively link their findings to existing studies and highlight the implications of their work. The manuscript adheres to ethical research and publication standards, and I have no concerns regarding dual publication or research ethics.

Overall, this is a high-quality manuscript that meets the standards of the journal. I highly recommend this work for subsequent processing and publication, subject only to any minor editorial adjustments that the journal may deem necessary. The authors are to be commended for their contribution, and I believe this paper will be of significant interest and value to the journal’s readership.

Reviewer #7: I noticed in this journal writing that you used PROSPERO, but after I checked, it was only at the initial screening stage up to completion. The same applies to the subsequent stages, which are the most important in PROSPERO, namely data extraction, bias/quality assessment, and data synthesis. Could you explain this further? Some of the arguments, results, and conclusions do not sufficiently elaborate on the analysis that demonstrates the novelty and urgency of this research.

Reviewer #9: Thank you for addressing issues and comments that have been provided before. It has been a great subject to focus on to do a research.

Reviewer #10: Review comments on “Virologic suppression among HIV-positive pregnant and lactating women receiving antiretroviral therapy in Africa: A systematic review and meta-analysis (PONE-D-25-49195)”

I have carefully reviewed the revised manuscript and the authors’ point-by-point responses. The authors have adequately and thoughtfully addressed all the comments raised. The PROSPERO registration has been done again and correctly clarified, redundancies in the introduction have been resolved, and methodological transparency has been substantially improved through the inclusion of a complete PRISMA 2020 checklist and a clearly presented flow diagram.

Importantly, the addition of meta-regression analyses strengthens the robustness of the findings, and the revised discussion now clearly aligns the results with the UNAIDS 95-95-95 targets and national PMTCT programs, highlighting strategies to improve viral suppression among younger women. Overall, the manuscript is scientifically sound, methodologically rigorous, and timely.

I have no further major concerns and recommend the manuscript for acceptance.

.

Reviewer #1: **Yes:** Salvatore ChirumboloSalvatore ChirumboloSalvatore ChirumboloSalvatore Chirumbolo

Reviewer #2: **Yes:** Hani Al-NajjarHani Al-NajjarHani Al-NajjarHani Al-Najjar

Reviewer #3: **Yes:** sri chakradhar challagalisri chakradhar challagalisri chakradhar challagalisri chakradhar challagali

Reviewer #6: **Yes:** James ArinaitweJames ArinaitweJames ArinaitweJames Arinaitwe

Reviewer #7: **Yes:** Janno Berty Bradly Bernadus, Sam Ratulangi University, Manado, IndonesiaJanno Berty Bradly Bernadus, Sam Ratulangi University, Manado, IndonesiaJanno Berty Bradly Bernadus, Sam Ratulangi University, Manado, IndonesiaJanno Berty Bradly Bernadus, Sam Ratulangi University, Manado, Indonesia

Reviewer #9: No

Reviewer #10: No

---

## [Author Response · Author response to Decision Letter 2]

12 Feb 2026

Dear Academic Editor and Reviewers,

We sincerely appreciate your prompt responses and for generously sharing your expertise through insightful comments and suggestions, which have been invaluable in guiding us to improve the quality of our manuscript. We have carefully addressed all comments to the best of our ability.

In response to the second-round revision, we have uploaded the revised manuscript with tracked changes, along with the updated main manuscript and the “Response to Reviewers” files.

---

## [Decision Letter · Decision Letter 2]

16 Mar 2026

Virologic suppression among HIV-positive pregnant and lactating women receiving antiretroviral therapy in Africa: A systematic review and meta-analysis

PONE-D-25-49195R2

Dear Dr. Mengistie,

We’re pleased to inform you that your manuscript has been judged scientifically suitable for publication and will be formally accepted for publication once it meets all outstanding technical requirements which include addressing some minor comments raised by one of the reviewers.

Kind regards,

Richard Makurumidze

Academic Editor

PLOS One

Additional Editor Comments (optional):

Reviewers' comments:

Reviewer's Responses to Questions

**Comments to the Author**

Reviewer #1: All comments have been addressed

Reviewer #7: All comments have been addressed

2. Is the manuscript technically sound, and do the data support the conclusions?

Reviewer #1: Partly

Reviewer #7: Yes

3. Has the statistical analysis been performed appropriately and rigorously?

Reviewer #1: No

Reviewer #7: Yes

4. Have the authors made all data underlying the findings in their manuscript fully available?

Reviewer #1: Yes

Reviewer #7: Yes

5. Is the manuscript presented in an intelligible fashion and written in standard English?

Reviewer #1: Yes

Reviewer #7: Yes

Reviewer #1: The revised manuscript demonstrates substantial improvement compared with earlier versions and addresses many of the reviewers’ concerns. In particular, the authors have strengthened the methodological transparency and analytical depth of the study. The inclusion of a meta-regression analysis to explore sources of heterogeneity represents an important improvement and directly responds to reviewer feedback requesting deeper analysis of variability among studies. This addition helps identify moderators such as ART regimen and geographic region, thereby improving the robustness and interpretability of the pooled estimates. The authors also expanded the discussion to better contextualize findings within socioeconomic and health-system factors such as rural–urban disparities, access to healthcare services, and ART adherence barriers. These additions enhance the explanatory power of the discussion and align the findings more clearly with public health policy frameworks such as the UNAIDS 95-95-95 targets.

Furthermore, the authors appear to have addressed comments regarding the background and the significance of the research topic. The revised manuscript provides a clearer justification of the study’s novelty and urgency, particularly by emphasizing the lack of continent-wide quantitative synthesis on virologic suppression among pregnant and lactating women in Africa. The manuscript now explicitly highlights gaps in evidence and the implications for prevention of mother-to-child transmission of HIV.

The reviewers also requested greater transparency regarding methodological procedures. In response, the authors clarified the PROSPERO registration and stated that data extraction, bias assessment, and synthesis were conducted according to the registered protocol. They also added the PROSPERO registration identifier and link in the manuscript, which improves methodological transparency.

However, not all concerns appear to be fully resolved. One reviewer noted that the manuscript previously reported an implausible study inclusion period extending to 2025, which could affect credibility. Although the authors provided clarification about the search timeframe and justification related to the SDG era, the explanation may still require clearer presentation in the manuscript. Additionally, transparency regarding quality appraisal remains partially limited; while the authors state that Joanna Briggs Institute scores are now included in supplementary materials, the main manuscript still lacks a concise summary of individual study quality.

Overall, the authors have fulfilled most of the reviewers’ comments, particularly regarding analytical depth, contextual discussion, and methodological clarification. Nevertheless, minor issues related to transparency and reporting clarity remain and should be carefully checked before final acceptance.

POINTS TO BE FURTER ADDRESSED

1. p values as 0.000 are meaningless and not scientific. You can use p <0.0001 or (better) report the correct p value as negative potencies (e.g, 5.78x10^-7);

2. Funnel plot. The funnel plot is not clearly correct/symmetric, and it suggests possible asymmetry, although it does not automatically prove publication bias.

Here is a brief scientific interpretation you could use: First, in a correct funnel plot, studies should be symmetrically distributed around the pooled effect size line, forming an inverted funnel. Larger studies (smaller standard error) should cluster near the top, while smaller studies should spread more widely at the bottom.

In the plot you reported, most studies cluster on the right side of the pooled estimate (~85%), while few studies appear on the left side. Additionally, several studies with larger standard errors (bottom of the plot) are predominantly located on one side. This creates visual asymmetry, meaning the funnel shape is not balanced. Such asymmetry may suggest small-study effects, publication bias, or heterogeneity among studies.

Another issue is the very wide spread of effect sizes (≈25–100%), which is unusual for prevalence meta-analysis and may reflect high heterogeneity, which is also consistent with the reported I² ≈ 99% in the manuscript. When heterogeneity is extremely high, funnel plots become less reliable for detecting publication bias.

However, visual inspection alone is insufficient. The correct interpretation should combine the funnel plot with statistical tests such as Egger’s regression test or Begg’s test. In the manuscript, Egger’s test was reported as p = 0.2402, indicating no statistically significant publication bias, even though the plot visually appears somewhat asymmetric.

Therefore, the appropriate conclusion is: the funnel plot shows some visual asymmetry, but according to the Egger’s test there is no statistically significant evidence of publication bias, and the asymmetry may be due to extreme heterogeneity across studies. This must be reported in the paper.

Reviewer #7: Thank you for revising all my review results. Please double check everything I provided and follow all the journal's obligations and requirements.

.

Reviewer #1: **Yes:** Salvatore ChirumboloSalvatore ChirumboloSalvatore ChirumboloSalvatore Chirumbolo

Reviewer #7: **Yes:** Janno Berty Bradly Bernadus, Sam Ratulangi University , ManadoJanno Berty Bradly Bernadus, Sam Ratulangi University , ManadoJanno Berty Bradly Bernadus, Sam Ratulangi University , ManadoJanno Berty Bradly Bernadus, Sam Ratulangi University , Manado

---

## [Editor Report · Acceptance letter]

PONE-D-25-49195R2

PLOS One

Dear Dr. Mengistie,

I'm pleased to inform you that your manuscript has been deemed suitable for publication in PLOS One. Congratulations! Your manuscript is now being handed over to our production team.

Kind regards,

on behalf of

Dr. Richard Makurumidze

Academic Editor

PLOS One